# Developing a cosmetic formulation containing lipase produced by the fungus *Aspergillus terreus*

**Gabriela Rocha Ramos[1], Elissa Arantes Ostrosky[2], Patrícia Santos Lopes[3], Newton Andréo Filho[3], Lohanna Luciyanla Kakuda[3], Jéssica Nascimento da Silva Pinto[3], Emanuela de Lima Viana[4], Neyna Santos Morais[5], Nathalia Saraiva Rios[6], Ana Paula Barreto Gomes[2], Francisco Humberto Xavier Júnior[7], Francisco Canindé de Sousa Júnior[1,2], Cristiane Fernandes de Assis[1,2]\***

1 Pharmaceutical Sciences Postgraduate Program, Center for Health Sciences, Federal University of Rio Grande do Norte, Natal, RN, Brazil, 2 Department of Pharmacy, Center for Health Sciences, Federal University of Rio Grande do Norte, Natal, RN, Brazil, 3 Pharmaceutical Sciences Postgraduate Program, Federal University of São Paulo, Diadema, SP, Brazil, 4 Nutrition Postgraduate Program, Center for Health Sciences, Federal University of Rio Grande do Norte, Natal, RN, Brazil, 5 Biotechnology Postgraduate Program, Federal University of Rio Grande do Norte, Natal, RN, Brazil, 6 Department of Chemical Engineering, Technology Center, Federal University of Rio Grande do Norte, Natal, RN, Brazil, 7 Department of Pharmacy, Center for Health Sciences, Federal University of Paraíba, João Pessoa, PB, Brazil

\* cristianeassis@hotmail.com, cristiane.assis@ufrn.br

## Abstract

This study aimed to evaluate the potential of lipase from *Aspergillus terreus* as an active ingredient in cosmetic formulations. Lipase was produced using the fungus *Aspergillus terreus* and was immobilized on gel silica as support. The enzymes were characterized using Scanning Electron Microscopy (SEM), Fourier Transform Infra-red Spectroscopy (FTIR), X-ray Diffraction (XRD), Thermogravimetry and Differential Scanning Calorimetry (TG/DSC), and safety evaluation through cytotoxicity tests using NIH-3T3 fibroblast cells. A central composite rotatable design was employed to find the best conditions for enzymatic cosmetic production. The enzyme produced by *A. terreus* showed activity of 375.9 U/g of substrate, and the immobilized enzyme showed 12.78 U/g of silica, while the lipase from *R. oryzae* showed activity of 69.91 U/g. As confirmed by FTIR and XRD, SEM showed weak enzyme interaction with silica during immobilization. Cytotoxicity tests showed that only the lipase produced by *A. terreus* was safe for NIH-3T3 fibroblast cells. The central composite rotatable design showed the agitation time influenced the enzyme activity response. According to the results, the enzyme produced by the fungus *A. terreus* is a promising and safe product for research into developing new cosmetic products.

## Introduction

Consumers increasingly perceive Globalization and industrial development as negative factors, especially those who purchase natural products free from synthetic

**Data availability statement:** All relevant data are within the manuscript and its Supporting Information files.

**Funding:** This study was partly financed by the Coordenação de Aperfeiçoamento de Pessoal de Nível Superior - Brasil (CAPES) - Finance Code 001. The funders had no role in study design, data collection and analysis, decision to publish, or preparation of the manuscript.

**Competing interests:** The authors have declared that no competing interests exist.

ingredients [1]. Given this scenario, the cosmetics sector has sought to respond quickly to the needs of these consumers [2]. In this context, using sustainable ingredients emerges as a beneficial approach by prioritizing natural raw materials in final products, respecting the properties of the skin and the environment, and, consequently, making a commitment to sustainability [3]. Microbial lipases constitute a significant group of enzymes due to their versatility and ease of production. These enzymes can catalyze hydrolysis, esterification, transesterification, interesterification, and aminolysis reactions in non-aqueous environments [4–6]. Furthermore, microbial lipases are highlighted by the ease of large-scale production, with submerged fermentation (SF) and solid-state fermentation (SSF) being the most effective techniques for their production [7–9].

Solid-state fermentation (SSF) is a technique that involves the growth of microorganisms using solid, porous, and moist particles as support, allowing cell growth and metabolism without exceeding the maximum water retention capacity of the matrix [10]. Among the fungi with the most significant evidence in lipolytic production, the genera *Geotrichum, Mucor, Penicillium, Rhizopus,* and *Aspergillus* stand out, with the last three being the most used in solid-state fermentation [11]. Aspergillus terreus, in particular, has stood out in the production of lipases and, in addition, is capable of producing biosurfactants and capable of carrying out esterification reactions, which makes it a promising option for industrial applications [12–14].

Lipases have characteristics that make them applicable in the cosmetics industry due to their ability to hydrolyze fats. They are commonly associated with shampoo and soap formulations for oily skin [15]. Such products can often act with other enzymes, helping remove dirt, fat deposits, and dead cells [16,17]. Some studies have also demonstrated the applicability of lipases in emulsions for reducing body fat [18]. These factors highlight the versatility of these enzymes, in addition to stimulating scientific interest in the development of innovative biotechnological products.

Enzyme immobilization has gained relevance, especially in the case of lipases, as it increases enzyme stability [19–22]. Immobilization protocols that use mesoporous silicas as support have been widely adopted as a laboratory technique, employing adsorption or covalent bonding processes [23–26]. This material has excellent potential for application in immobilization, mainly due to the possibility of modifying its surface, which offers thermal and mechanical stability and toxicological safety, increasing the chances of implementing these enzymes in various industrial scenarios [27,28].

Considering the biotechnological potential of lipases, this study aimed to investigate a commercial lipase from *Rhizopus oryzae* (LC) and to produce a lipase from *Aspergillus terreus* (LP) by solid-state fermentation, testing the immobilization of the latter enzyme in silica gel by physical adsorption. In addition, the project aimed to evaluate the *in vitro* safety of both enzymes, in free and immobilized forms, to allow their incorporation into stable cosmetic products with efficacy and safety, thus offering promising methods for scientific and technological development in the pharmaceutical and cosmetic areas.

## Materials and methods

### Commercial lipase

The commercial lipase was kindly donated by UNIFESP, Diadema-SP campus, produced by the company Nagase Chem-teX Corpoation of *Rhizopus oryzae* (Lilipase A-10D – lot 2035157, with enzyme patent 20110053229).

### Lipase enzyme production

The *Aspergillus terreus* (NRRL-269) microorganism was provided by the "ARS Culture Collection" microorganism bank (Peoria – Illinois, USA). *Aspergillus terreus* was activated on plates containing PDA (Potate Dextrose Agar) medium and incubated at 30°C for 7 days. The plates were maintained in PDA medium at 4°C and subcultured every month.

Solid state fermentation was performed using 9 g of wheat bran and 1 g of extra virgin olive oil inducer. A salt solution was prepared to supplement the medium with $NaNO_3$ and 0.5% (w/v) $KH_2PO_4$; yeast extract 0.2% (w/v); $MgSO_4.7H_2O$ and KCl 0.1% (w/v) and $FeSO_4.7H_2O$, $ZnSO_4.7H_2O$, MnCl and $CuSO_4.5H_2O$ 0.001% (w/v). The flasks were inoculated with $1 \times 10^7$ spores/g of solid substrate and 7 mL of the salt solution was added. The vials containing this mixture were incubated at 37°C and kept for 120 hours [16].

### Enzyme extraction

Crude enzymatic extraction was conducted with 50 mL of Tris-HCl buffer (0.05 M, pH 8) into the fermented medium. The mixture was maintained at 30° C in rotatoty shaker (QUIMIS, Q816M20, Diadema, Brazil) at 200 rpm for 30 minutes. After extraction, the content was filtered through filter paper and centrifuged at 3,000 rpm for 10 minutes. The supernatant (crude enzymatic extract) was used to determine lipolytic activity [16].

### Determination of lipolytic activity

The supernatant was used for the hydrolysis of the chromogenic substrate p-nitrophenyl palmitate – pNPP [29]. Enzyme activity was evaluated by adding 10 µL of enzymatic extract and 90 µL of pNPP emulsion, and incubated at 37°C for 30 min. The amount of liberated p-nitrophenol was recorded at 405 nm. A calibration curve ($R_2 = 0.9987$) was built with concentrations from 4.2 to 140.0 mmol of p-nitrophenol. A unit of lipolytic activity was defined as the amount of p-nitrophenol released per minute per reaction volume (mmol/min/mL).

### Determination of total proteins by the Bradford method

For protein determination, the Bradford method was used [30]. 100 µL of sample were placed in test tubes, and 2.5 mL of Bradford reagent was added. After 5 minutes of reaction, the reading was performed on a UV-VIS spectrophotometer (Biospectro, Mod. SP-220) at 595 nm. The results were calculated according to a standard curve of bovine albumin (0.3 to 2 mg/mL).

### Immobilization of lipase by physical adsorption using silica gel as support

A concentration of 1 mg of enzyme/g of support was used in 3.85 mL of crude enzymatic extract (obtained according to item 4.2) for 58.158 mL of 100 mM Tris-HCl buffer pH 7.0 (immobilization solution). The blank was considered as 2 mL of the immobilization solution. Then, the 60 mL of the immobilization solution was placed in contact with the silica. The materials were constantly stirred for 24 hours at room temperature in a tube homogenizer. After 24 h, the immobilized material was left to rest until maximum decantation of the support (silica), the supernatant was separated into 15 mL falcon tubes and centrifuged, and the centrifugation supernatant was collected. The silica was washed using 100 mM Tris-HCl buffer, pH 7.0, in sufficient quantity to make the precipitate light in color. Then, it was filtered using a vacuum pump (brand) to determine the enzymatic activity. The enzymatic activity was measured for the immobilized enzyme, 24-hour blank, and

centrifugation supernatant following the methodology described in topic 4.2. The Bradford analysis was performed for the 24-hour blank and centrifugation supernatant. The immobilization yield was calculated using Eq 1:

$$Yield = \frac{\text{Immobilized enzyme activity}}{\text{Initial enzyme activity}} \; x \; 100$$

(1)

At the end of the assay, the immobilized lipase was stored in Eppendorf tubes at a temperature of – 4 °C.

## Characterization of free, immobilized lipase and commercial lipase

**Scanning electron microscopy (SEM).** The samples were inserted into a carbon tape fixed to a metal support (stubs) and then metallized with a thin layer of gold. The analyzes were carried out in high vacuum, voltage equivalent to 30 mA/min, with the images accelerated to a voltage of 15 kV and captured at 100, 250 and 500x magnification on a Hitachi Tabletop TM-3000 Microscope.

**Fourier transform infrared spectroscopy (FTIR).** Free, immobilized, and commercial lipase samples were homogenized separately with potassium bromide (KBr). They were then macerated and pressed to obtain pellets. The spectra were recorded in transmittance with the mid-infrared region (400–4000 cm$^{-1}$). A Shimadzu FTIR-8400S IRAFFINITY-1 series spectrometer and the IRSOLUTION version 1.60 program with a scan number of 32 and a resolution of 4 cm$^{-1}$ were used.

**X-ray diffraction (XRD).** The samples were placed in a cylindrical sample holder and analyzed at a diffraction angle of 2ꝋ between 0 and 100° using a high-resolution X-ray diffractometer (SHIMADZU XRD 7000) with a Seifert ID3000 generator.

**Thermogravimetric analysis and differential scanning calorimetry (TG/DSC).** This analysis was performed using the simultaneous thermal analysis methodology, thermogravimetric analysis (TGA) and Differential Scanning Calorimetry (DSC) simultaneously on a Discovery SDT-650 device (TA Instruments, USA) with sealed alumina crucibles. A constant heating rate of 10 °C.min$^{-1}$, nitrogen air flow of 50 mL.min$^{-1}$ and a temperature range of 25°C to 450°C were used. Graphics were integrated using the TRIOS program.

## Assessment of the cytotoxicity of free lipase, immobilized lipase and commercial lipase

The cell viability of NIH/3T3 cells (CRL – 1658) was evaluated at different sample concentrations (0.78 to 100 μg/mL) according to the OECD methodology [31]. Cells were plated in 96-well plates at a concentration of approximately 1x10$^5$ cells/mL, including a control group. The plates were incubated to allow cells to adhere for 24 hours. Then, different concentrations of free lipase, immobilized lipase and commercial lipase samples were added to the cells and incubated for another 24 hours. After this period, the microplates were washed, followed by neutral red dye application. After the final wash, microplates were read at 540 nm using a microplate reader. The calculation of cell viability was performed according to Eq 2.

$$Reduction\ of\ NR = \frac{\text{Abs sample}}{\text{Abs control}} \; x \; 100$$

(2)

## Characterization of free *A. terreus* lipase

**Influence of temperature and pH on enzyme activity.** The effects of temperature and pH were performed using the crude enzymatic extract. The temperature ranged from 30 to 80°C, while the pH ranged from 3.0 to 10.0. The buffers used were phosphate-citrate (50 mM, pH 3.0 to 7.0), Tris-HCl (0.05 M, pH 8.0), and sodium bicarbonate buffer (pH 9.0 and 10.0). The relative activity was calculated at pH 7.0 and 37º C as a control.

**Influence of compounds on enzyme activity.** The influence of metal ions and chelating agents (10 mM) on lipolytic enzymatic performance was tested for: NaCl, $K_2SO_4$, $CaCl_2$, $CaSO_4 \cdot 7H_2O$, $CuSO_4 \cdot 5H_2O$, $MgSO_4 \cdot 7H_2O$, $MnSO_4 \cdot 7H_2O$, $ZnSO_4 \cdot 7H_2O$, $FeSO_4 \cdot 7H_2O$, $NiCl_2 \cdot 6H_2O$, EDTA, Triton x-100 and sodium docetyl sulfate (SDS). All experiments were carried out at 37°C for 30 minutes and enzymatic activity was performed according to item 2.4. The results were expressed as relative activity (%), using 100 as a control.

## Experimental planning of cosmetic formulations containing *Aspergillus terreus* lipase

A $2^2$ central composite rotatable design (CCRD) was used with a duplicate at the central point to define the best conditions for preparing cosmetic formulations containing *A. terreus* lipase. The parameters used as independent variables (factors) were enzyme concentration and stirring time and the dependent variables (responses) were pH, droplet size, Zeta potential, polydispersity index (PDI) and enzyme activity (Table 1).

The raw materials used to obtain the cosmetic formulation, named according to the International Nomenclature of Cosmetic Ingredient (INCI), were: Lipase from *A. terreus*, Ethylhexylglycerin and Phenoxyethanol, *Aspergillus terreus* produced lipase and glycerin, and purified water. Table 2 shows the qualitative and quantitative description of the manipulated cosmetic emulsion. The experimental planning to obtain the cosmetic formulation containing lipase generated 11 different experimental concentration and stirring conditions.

The emulsifying base used [32] enabled obtaining a cold emulsion with stirring carried out in an ultra-turrax (IKA, mod. T18) 11000 rpm for times pre-determined by experimental analysis. The enzymatic activity for each gram formulation corresponds to 7.518 U/g

**Zeta potential.** Para a análise do potencial zeta da formulação cosmética, 1g de cada amostra foi diluída em 9g de água destilada e colocadas em cubetas específicas. Foram realizadas dez execuções em 1 minuto, e foi utilizado o software *Zetasizer (Malvern Instruments Ltd) - modelo Nano ZS.*

Ten executions were performed in 1 minute, and the Zetasizer Nano ZS model software (Malvern Instruments Ltd) was used.

**Dynamic light scattering (DLS).** To determine the particle size, 1g of each sample was diluted in 9g of freshly distilled water and vortexed for 30 seconds. The average particle size was evaluated using the Zetasizer (Malvern Instruments Ltd) equipment—a Nano ZS model with a detection angle of 173º. The equipment has a monochromatic laser

**Table 1. Factors and levels used in the central composite rotational design (CCRD) $2^2$ for cosmetic formulations using lipase from *Aspergillus terreus*.**

|  | Levels |  |  |  |  |
|---|---|---|---|---|---|
| Factors | -1.41 | -1 | 0 | +1 | +1.41 |
| Enzyme concentration (m/v %) | 8.96 | 10 | 12.5 | 15 | 16.03 |
| Agitation time (min.) | 3.96 | 5 | 7.5 | 10 | 11.03 |

**Table 2. Qualitative and quantitative description of the manipulated cosmetic emulsion.**

| Components | Concentration (%w/w) | Quantity for 50g |
|---|---|---|
| Plantus 360˚ | 2.0 | 1g |
| EDTA | 0.1 | 0.05g |
| Glycerin | 3.0 | 1.5g |
| Proteg SL˚ | 0.5 | 0.25g |
| Lipase from A. terreus Distilled water | q.s.p<br>q.s.p 50g | q.s.p<br>q.s.p |

with a wavelength of 673 nm. Measurements were performed in triplicate (60 s for each measurement). The data were analyzed using the NANO-flex Control 0.9.7 program. The entire experiment was carried out in triplicate.

**Measuring the pH of cosmetic formulations.** The pH value was determined using a benchtop pH meter (LUCA-210, Tecnopon).

**Determination of cosmetic formulation enzyme activity.** First, 2.5 mL of 0.05M TRIS-HCl buffer pH 8.0, 1 mL of olive oil and 2 mL of 2% polyvinyl alcohol were added in a 250 mL Erlenmeyer flask. The material was incubated in an orbital shaker at 37°C and 150 rpm for 1 minute, and then 1 mL of the sample (crude enzymatic extract) was added. The reaction was stopped with a 1:1 acetone and ethanol solution. Titration was performed with 0.05 M NaOH until a pink color was obtained. The control was made using water instead of enzyme. One lipase unit (U) was defined as the amount of enzyme that releases 1 mmol of fatty acids under the assay conditions described above. Calculations for enzymatic activity were performed according to eq 3:

$$Enzyme\ activity\ (U/mL) = (Va - Vb) x\ 50\ x\ 0.1$$

(3)

In which: Va is the volume of NaOH titrated into the sample; Vb is the volume of NaOH titrated into the blank, 50 is a fixed value for the calculation and 0.1 is the dilution factor corresponding to the samples [32,33].

### Statistical analysis

Data were expressed as mean and standard deviation, and analyzed using the STATISTICA version 7.0 program. The normality of data distribution was initially assessed using the Kolmogorov-Smirnov test. In case of normal distribution, data were analyzed by parametric tests using ANOVA followed by Tukey's post-test. Non-parametric data were analyzed using the Kruskal-Wallis test and Dunn's post-test. A $p$-value $< 0.05$ was adopted for all analyzes to determine statistically significant differences. Central composite rotatable design (CCRD) experiments were performed randomly. The statistical significance of the second-order model equation was determined by the F test (ANOVA).

## Results and discussion

### Evaluation of enzyme activity

The production of lipases stands out due to their mass production and good versatility, with filamentous fungi considered good enzyme producers [8,34,35]. This study chose to use the filamentous *Aspergillus terreus* fungus to produce lipase from solid-state fermentation, using wheat bran as a substrate and extra-virgin olive oil as an inducing agent, as this type of fermentation provides good yields and low cost when compared submerged fermentation [36].

The commercial *Rhizopus oryzae* lipase showed an enzymatic activity of 69.91 U/g. The results found for this study showed that the *Aspergillus terreus* lipase enzyme produced presented an enzymatic activity of 375.9 U/g of residue used for fermentation. Azevedo et al. (2020) [16] found similar results using olive oil as an inducer for lipase production from *Aspergillus terreus* (303.9 U/g), while Barros et al. (2023) [12] found activity of 70.1U/g using Bati butter as an inducing agent for lipase enzyme production by *A. Terreus*.

### Enzyme immobilization by physical adsorption using silica

Among the methods that aim to increase enzyme stability, enzyme immobilization has gained relevance when dealing with lipases, conferring increased stability and better productivity in enzymatic processes [19–23]. Enzyme immobilization protocols that use mesoporous silicas as support have been widely chosen as a laboratory technique involving adsorption or covalent binding processes [24–26]. The result of enzymatic activity found after the enzymatic immobilization of lipase produced from *Aspergillus terreus* was 12.78 U/g of silica and the yield was approximately 26.5%. Adsorption immobilization in this type of enzyme immobilization, enzymes are immobilized on the support through bonds such as hydrophobic

interactions, Van der Waals forces, hydrogen bonds, and ionic bonds. The low immobilization yield can be explained due to the weak physical adsorption interaction, which may have led to the enzyme leaching in the reaction media [27,36,37]. Furthermore, the adsorption efficiency of an enzyme on the surface of a support is related to several parameters such as protein size, surface area of the adsorbent, porosity, pore size and enzyme concentration [38,39]. According to the results, further studies or the use of other types of silica are necessary to increase immobilization efficiency.

## Physical-chemical characterization of free, immobilized and commercial lipase

**Infrared spectroscopy with fourier transform (FTIR).** Fig 1 presents the spectra obtained for silica gel, immobilized *Aspergillus terreus* lipase, free *Aspergillus terreus* lipase and commercial *Rhizopus oryzae* lipase.

The results referring to free *Aspergillus terreus* lipase (Fig 1, line c) and commercial *Rhizopus oryzae* lipase (Fig 1, line d) showed the presence of bands 1636 cm$^{-1}$ and 1640 cm$^{-1}$, respectively, corresponding to the primary and secondary amino groups, characteristic of lipases [40]. The same band was evident in the spectrum of the immobilized *Aspergillus terreus* lipase (Fig 1, line b), also demonstrating the presence of the enzyme in the sample supported for immobilization. Furthermore, the three types of enzymatic samples showed the presence of bands varying between 3295 − 3328 cm$^{-1}$ corresponding to alcohols and phenols, as well as the presence of the C − O group, represented by bands that varied from 1000 − 1056 cm$^{-1}$ [41].

The presence of peaks at 799 cm$^{-1}$ and 975 cm$^{-1}$ was highlighted when analyzing the spectra obtained for silica gel (Fig 1, line a), being characteristic of siloxane groups (Si-O-Si) [42,43]. In comparing the peaks obtained for silica gel with each of the lipase samples, it is noted that vibrational bands varying in the range of 784 − 799 cm$^{-1}$ and 971 − 975 cm$^{-1}$ were present in lipase immobilized with silica gel, which proves that there was interaction of both enzymes with silica gel.

**Scanning electron microscopy (SEM).** Fig 2A–2C show the micrographs corresponding to silica gel at a magnification of 100x, 250x and 500x, respectively, while Fig 2D and 2E show the micrographs corresponding to immobilized *Aspergillus terreus* lipase and commercial *Rhizopus oryzae* lipase.

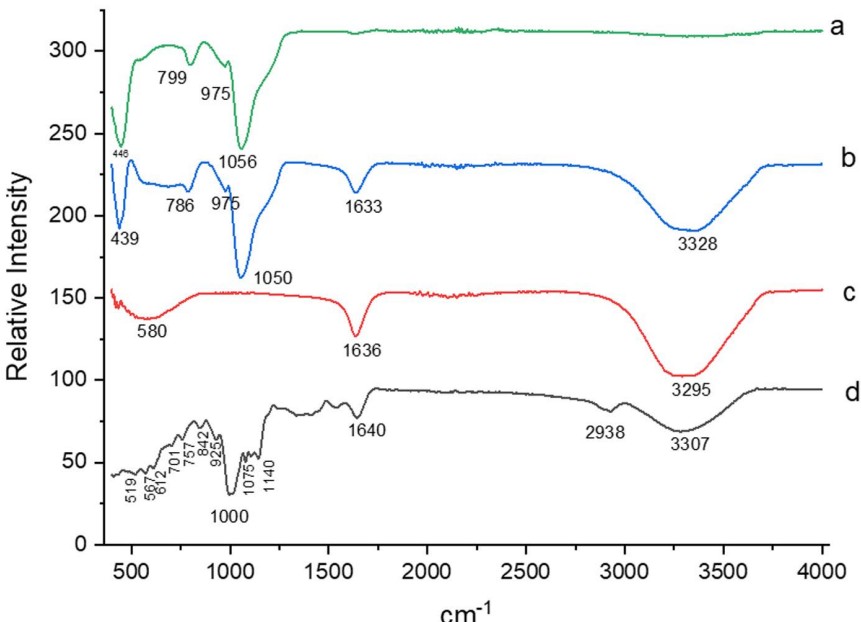

**Fig 1. Fourier Transform Infrared Spectroscopy (FTIR): (a): Silica gel; (b): *Aspergillus terreus* lipase produced immobilized; (c): *Aspergillus terreus* lipase produced free; (d): commercial lipase from *Rhizopus oryzae*.**

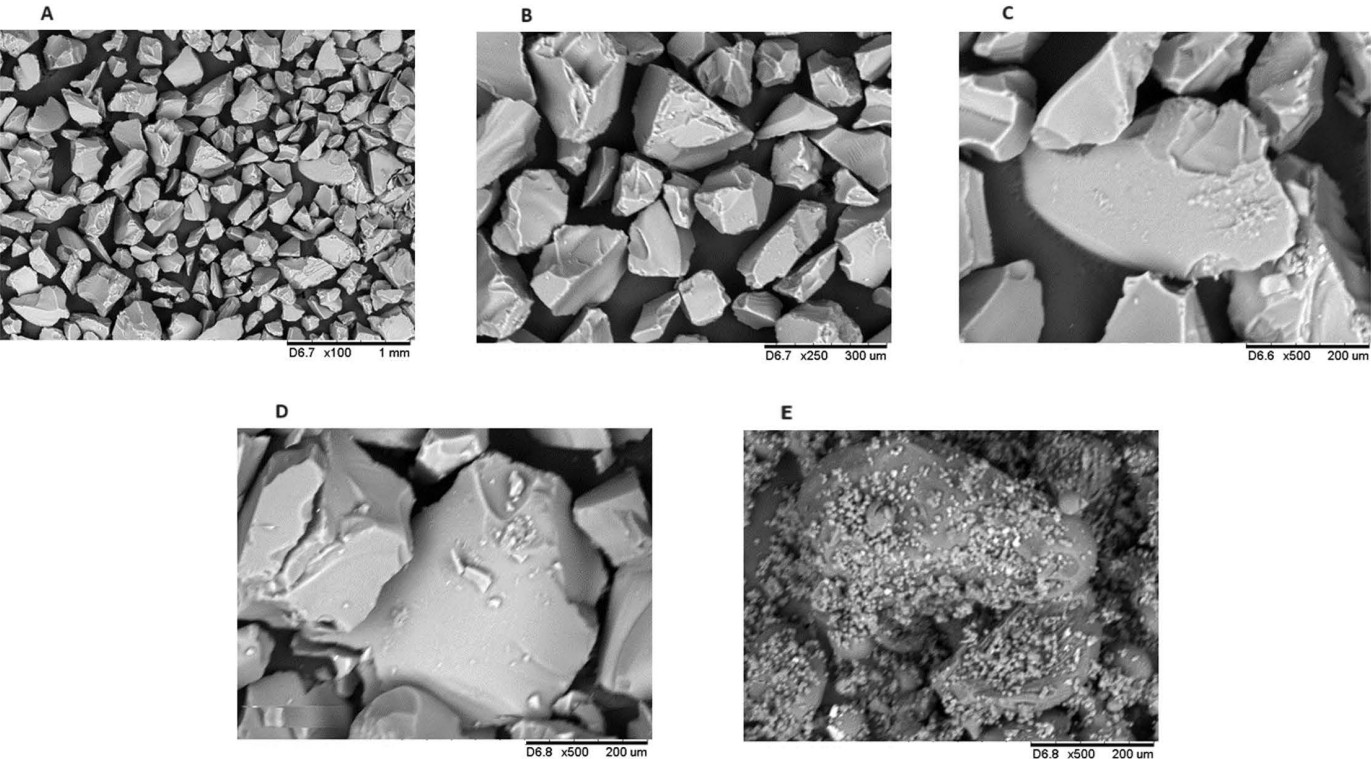

**Fig 2. Scanning Electron Microscopy for Silica Gel and Lipases: (A) silica gel magnification 100x; (B) silica gel magnification 250x; (C) silica gel 500x magnification (D) immobilized lipase from *Aspergillus terreus* produced 500x magnification; (E) commercial lipase from Rhizopus oryzae magnification 500x.**

When analyzing the scanning electron microscopy (SEM) images referring to the morphology of the silica gel, it is possible to notice particles that are mostly uniform (100x magnification). Then, brief visualization of the particles' surface can be seen by increasing the micrograph magnification (250x), which also appears to be uniform. Next, 500x magnification enabled visualizing a rigid structure in a single block, with apparent roughness and depressions on its surface, similar to the pure silica micrograph corresponding to the study carried out by [44]. This result is considered satisfactory for the use of silica as a support for immobilization, as the grooves present on the surface of the material favor enzymatic adsorption due to the increase in the surface area available for interactions with enzymes [45].

Fig 2D and 2E provide a microscopic analysis comparison for immobilized *Aspergillus terreus* lipase and commercial *Rhizopus oryzae* lipase. Microscopy referring to the immobilized *A. terreus* lipase did not show visually relevant differences when compared to the pure silica gel image, which could represent a sign that there was not good immobilization, as the interaction by physical adsorption is weak and can cause enzyme leaching in the reaction media, requiring further studies related to the porosity influence of the support in this method, as well as the size of the lipase to be immobilized [27]. Furthermore, it was noted that the commercial lipase sample showed visually relevant differences when compared to the silica gel micrograph. The image for commercial *R. oryzae* lipase showed rounded surfaces, which may refer to the support for immobilization, with granulations on its surface, which suggest they are enzyme aggregates.

**X-ray diffraction (XRD).** Fig 3 shows the diffractograms referring to silica gel, immobilized *Aspergillus terreus* lipase with silica gel and commercial *Rhizopus oryzae* lipase. When analyzing the diffractograms obtained, it was possible to observe that the silica gel (A) presents amorphous characteristics with 2θ = 15° to 30° [46–48]. An amorphous region with very subtle crystalline peaks was observed in the peaks of the commercial *Rhizopus oryzae* lipase (B),

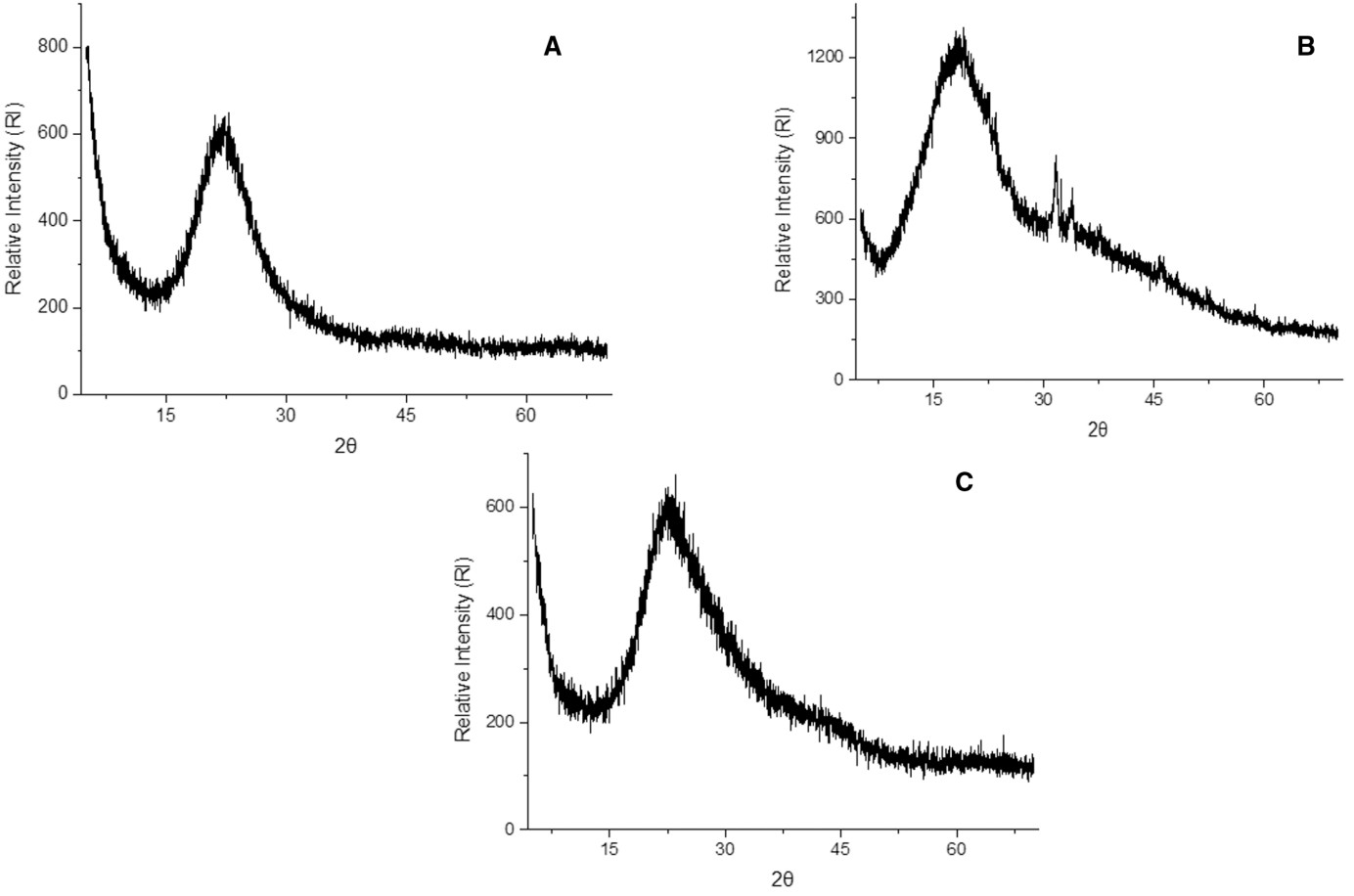

**Fig 3. X-ray diffraction: (A) silica gel; (B) immobilized *Aspergillus terreus* lipase; (C) commercial lipase from *Rhizopus oryzae*.**

while the immobilized *Aspergillus terreus* lipase presented an amorphous structure similar to silica, confirming that the immobilization possibly did not occur satisfactorily, reinforcing the results visualized in SEM.

**Analysis of thermogravimetry and differential scanning calorimetry (TG/DSC).** The TG and DSC curves for silica gel, free *A. terreus* lipase, immobilized *A. terreus* lipase and commercial *Rhizopus oryzae* lipase are represented in Fig 4A and 4B.

The silica mass loss (Fig 4A) was approximately 0.3% up to a temperature of 450°C. A study conducted by Souza et al. (2013) [40] found values of 22% mass loss of pure silica, however in analysis with temperatures up to 1000°C. The silica mass loss can be attributed to the presence of non-reactive silanol groups of the silane precursor present in silica due to incomplete sol-gel reactions [49]. Furthermore, mass loss can also occur due to the removal of water molecules that were strongly bound to the silica matrix [50]. The free *A. terreus* lipase (Fig 4A) begins to decompose at 28°C up to a temperature of 103°C, with a mass loss of 99%. Immobilized *Aspergillus terreus* lipase presents greater stability regarding mass loss, which is probably related to silica, with a loss of 44% occurring up to 89°C (Table 3).

This result may be mainly associated with the water extraction from the surface and decomposition of amino groups, generally organic groups [26,40].

Therefore, the lower values obtained for mass loss associated with immobilized lipase are the result of greater thermal stability of the matrix from interactions between silane precursors and lipase, as observed by Soares et al [26].

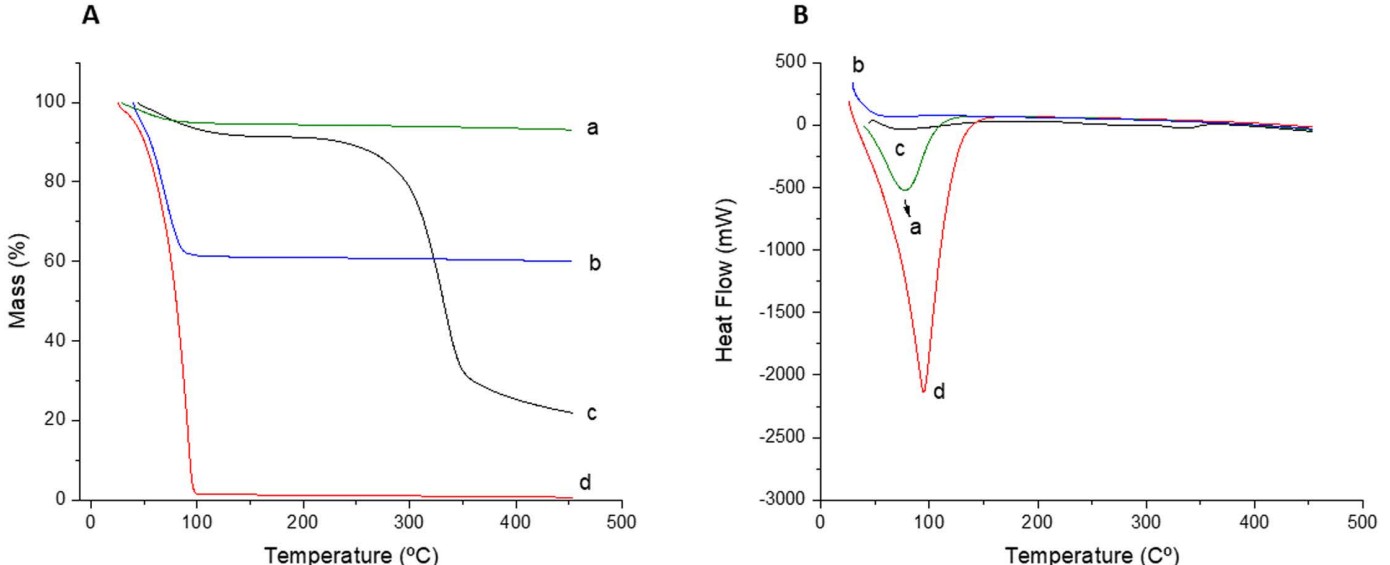

**Fig 4. (A)** Thermogravimetry (TG) curves and **(B)** Differential Scanning Calorimetry (DSC) curves: (a) Silica gel, (b) immobilized lipase from *Aspergillus terreus* (c) commercial lipase from *Rhizopus oryzae*; (d) free *Aspergillus terreus* lipase.

**Table 3. Results obtained through Thermogravimetry (TG) for silica, lipase enzyme, immobilized lipase, produced lipase, immobilized produced lipase.**

| Samples | 1ª stage | | | 2ª stage | | | 3ª stage | | | Total weight lossl/% |
|---|---|---|---|---|---|---|---|---|---|---|
| | *Ti/oC* | *Tf/oC* | Weight loss/% | *Ti/oC* | *Tf/oC* | Weight loss/% | *Ti/oC* | *Tf/oC* | Weight loss/% | |
| Silica | 31 | 101 | 0 | – | – | – | – | – | – | 0 |
| Commercial lipase | 46 | 118 | 8 | 237 | 363 | 62 | 363 | 450 | 10 | 80 |
| Immobilized lipase | 24 | 89 | 44 | – | – | – | – | – | – | 44 |
| Free lipase | 28 | 103 | 99 | – | – | – | – | – | – | 99 |

Commercial *Rhizopus oryzae* lipase shows gradual mass loss in three stages, as shown in Table 3, with a more substantial reduction of 62% occurring up to 363°C, also showing greater stability than immobilized lipase in terms of decomposition temperature.

The sample containing the free *A. terreus* lipase (Fig 4B) showed a first endothermic peak with an onset temperature of 27°C and a peak of 95°C, with high enthalpy (47,447kJ/g), associated with the decomposition of organic matter and water loss (Table 4).

The pure silica sample shows a small endothermic peak around 50°C (Enthalpy - 5947KJ/g), constituting a similar result to that found by Souza et al. (2013) [40]. The immobilized *A. terreus* lipase showed an endothermic peak at 75°C (Enthalpy – 23114KJ/g), and the commercial *R. Oryzae* lipase also showed an endothermic peak at 75°C, but with a lower enthalpy for the first endothermic transition in relation to immobilized *A. terreus* lipase (Enthalpy - 2890KJ/g). This shows that the type of origin of the lipase influences its stability.

## Cytotoxicity evaluation of free, immobilized and commercial lipase

With the aim of incorporating the lipase enzyme into a cosmetic formulation, this test aims to evaluate cytotoxicity using fibroblast cells, which are responsible for playing a fundamental role in maintaining the integrity and homeostasis of connective tissue, being the main cells involved in the tissue repair process [51].

**Table 4. Results obtained through Differential Scanning Calorimetry (DSC) for silica, lipase enzyme, immobilized lipase, produced lipase, immobilized produced lipase.**

| Samples | Transitions | Temperature/oC | | Energy (J.g-1) |
|---|---|---|---|---|
| | | Onset | Peak | |
| Silica | 1 | 29 | 50 | 5947 |
| Commercial lipase | 1 | 49 | 75 | 2890 |
| | 2 | 229 | 261 | 294 |
| | 3 | 307 | 337 | 601 |
| Immobilized lipase | 1 | 23 | 75 | 23114 |
| Free lipase | 1 | 27 | 95 | 47447 |

The *Aspergillus terreus* lipase (Fig 5A and 5B) did not show cytotoxicity to fibroblast cells compared to the control. Cell viability was 100% at the different concentrations tested.

The commercial *Rhizopus oryzae* lipase (Fig 5C) showed that there was a significant reduction in cell viability ($p > 0.05$) in the neutral red assay compared to the control at all tested concentrations. Cell viability decreased by approximately 80.31%, representing a statistically significant reduction, indicating that the commercial *R.oryzae* lipase is not safe in fibroblast cells at these concentrations.

Silica gel was used as a support for immobilization and the results of its cytotoxicity are shown in Fig 5D. By statistically comparing the cell viability results for the silica gel concentrations tested and the cell control, it was possible to conclude that the material decreased the viability of NIH-3T3 fibroblast cells by approximately (35.73%). According to Balduzzi et

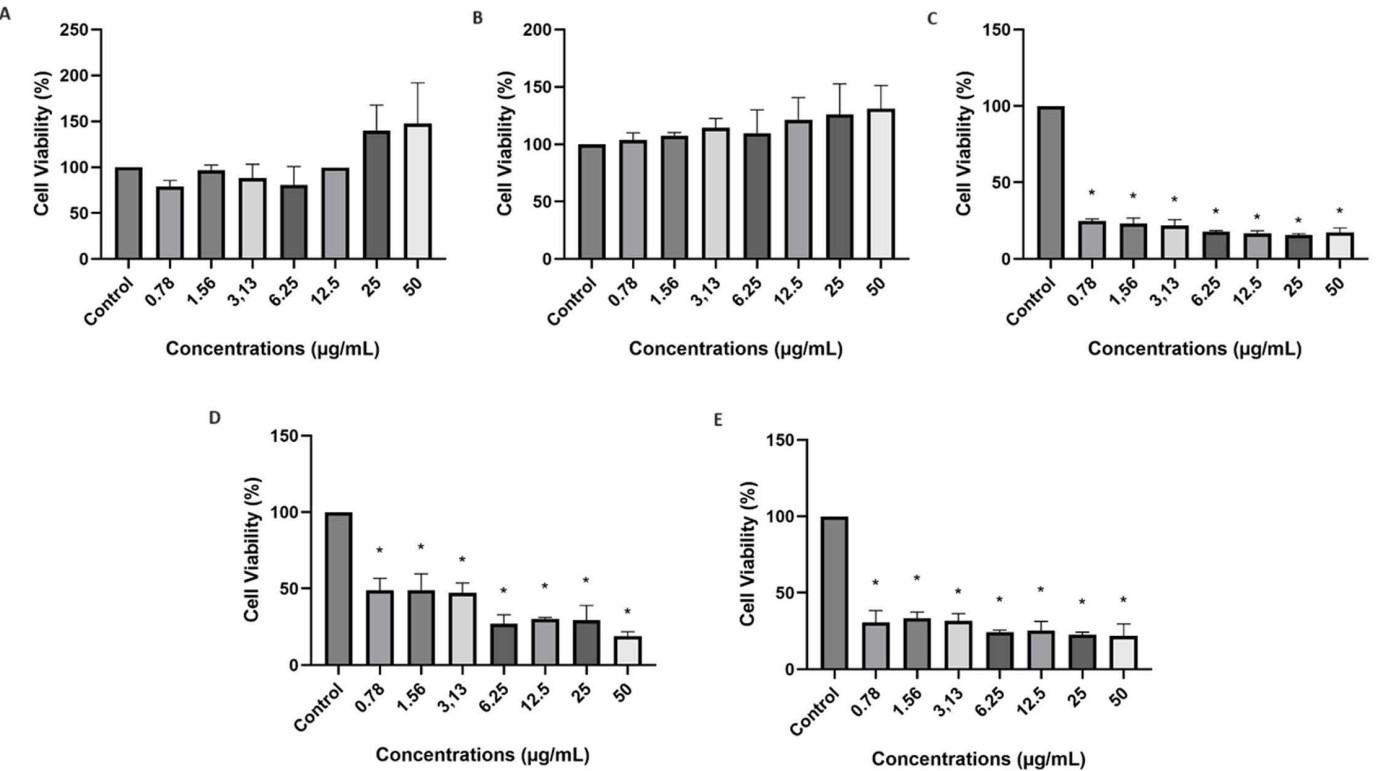

**Fig 5. Assessment of Cellular Viability of NIH-3T3 Fibroblasts.** (A) free *Aspergillus terreus* lipase dissolved in D10 medium, (B) free *Aspergillus terreus* lipase dissolved in ethanol, (C) commercial lipase from *Rhizopus oryzae*, (D) silica gel, (E) immobilized *Aspergillus terreus* lipase. * Statistically significant difference ($p < 0.05$) of the control using one-way ANOVA followed by Tukey post-test.

al. (2004) and Razzaboni and Bolsaitis (1990) [52,53], most *in vitro* studies using crystalline silica have demonstrated that this material is quite cytotoxic, which may be related to the presence of highly reactive radicals on the surface of these particles which act on cytoplasmic membrane and induce lipid peroxidation and protein denaturation.

Therefore, the results for immobilized *Aspergillus terreus* lipase also did not show safety for use in NIH-3T3 fibroblast cells (Fig 5E). The viability for immobilized *Aspergillus terreus* lipase was approximately 27.13%. At the end of this test, the free *Aspergillus terreus* lipase presented the best safety conditions for use in a cosmetic formulation, and was therefore chosen to continue the following tests. This study is a pioneer in evaluating the cytotoxicity of lipase from *Aspergillus terreus*.

### Characterization of free *Aspergillus terreus* lipase

**Effect of temperature and pH on lipase enzyme activity.** The characterization of an enzyme in terms of its ability to adapt to different external conditions directly influences its functionality; therefore, studying the influence of temperature and pH is essential to avoid protein denaturation and consequently the loss of its activity.

The results regarding the influence of temperatures on the enzymatic activity of free *Aspergillus terreus* lipase are shown in Fig 6A. The statistical results at the end of the test showed a significant reduction in enzymatic activity at all temperatures (40°C, 60°C, 70°C, 80°C and 100°C) when compared to the control temperature (30°C), with 40°C being the temperature that least interfered with lipolytic performance. Souza et al. (2019) [54] studied lipases from filamentous fungi and found that the optimal temperature for enzymatic activity ranged from 25°C to 45°C.

Furthermore, the effect of pH is an important parameter in the stability of enzymatic activity, as a small variation can reduce its activity due to influences on the conformation of the catalytic site, and extreme changes can completely alter the enzyme structure and lead to its denaturation [16,55]. Evaluating the results found in Fig 6B, it was possible to conclude that there was a significant influence on the enzymatic activity for all tested pHs (3.0 to 10.0) when compared with the control (100mM Tris-HCl buffer pH 7.0). When analyzing the influence of pH on the enzymatic performance of *Aspergillus terreus* lipase, Azevedo et al. (2020) [16] concluded that the best pH for lipolytic activity would be 7.0.

**Influence of ions, chelating agents and surfactants on enzymatic activity.** Metal ions have the ability to interfere with the enzymatic structure, binding to enzymes, being able to modify their conformation and alter activity and/or stability [56–58]. Therefore, the results of the analysis of metal ions and chelating agents are shown in Fig 7.

When analyzing the results, it is noted that there were significant differences for enzymatic activity when in contact with almost all ions, except NaCl and $K_2SO_4$. The lipase reading performed under the influence of Calcium Chloride ($CaCl_2$) increased

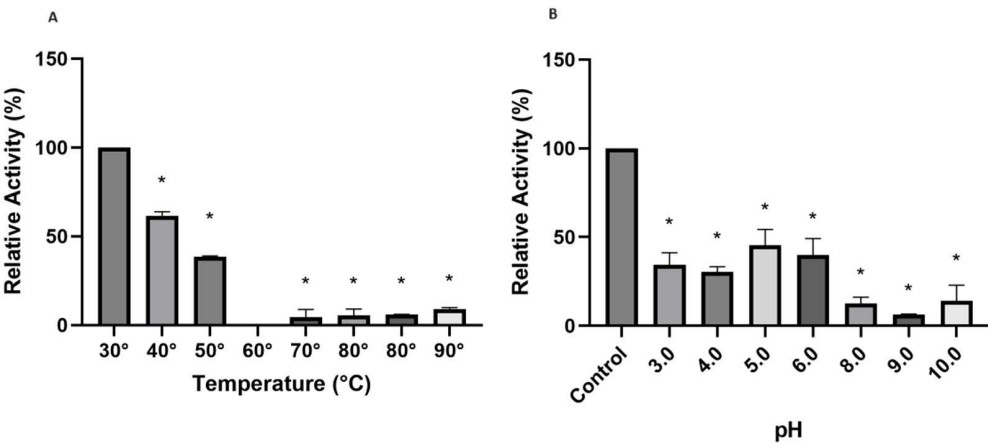

**Fig 6. Relative activity at different temperatures (A) and pH (B).** *Statistically significant difference ($p < 0.05$) of the control using one-way ANOVA followed by Tukey post-test.

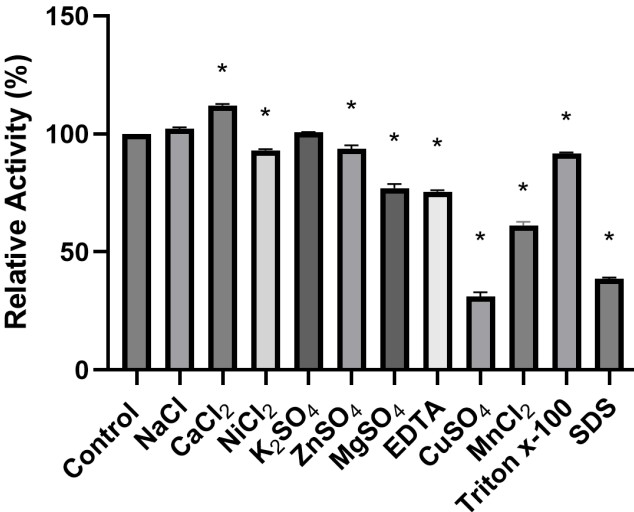

**Fig 7. Relative activity in different metal ions, EDTA and surfactants.** *Statistically significant difference (p < 0.05) of the control using one-way ANOVA followed by Tukey post-test.

significantly in relation to the control, showing higher enzymatic activity than the control, which may be indicative that this ion behaved as an enzymatic cofactor in this case [59,60]. On the other hand, the sulfate ion ($CuSO_4$) and the surfactant docetyl sodium sulfate (SDS) showed a significant reduction in activity, but acting as reducers of enzymatic activity. In studies carried out by Lima et al. (2004) [61], it was identified that $CuSO_2$ also acted to reduce enzymatic activity. Furthermore, the use of surfactants is commonly used in the production of emulsions with the purpose of detecting lipolytic activity; however, their use can cause an increase in enzymatic performance or cause a change in the structure of the enzyme, leading to its denaturation [62].

### Experimental design of cosmetic formulation containing lipase enzyme from Aspergillus terreus

An experimental design was implemented in order to identify the best experimental conditions to develop an enzymatic cosmetic formulation. The results obtained in the rotational central composite design are shown in Table 4.

Depending on the experimental condition adopted, it is noted that the droplet size response varied from 443.5 nm to 1274.3 μm, the PDI varied from 0.704 to 0.900, the zeta potential ranged from -55.93 to -70.6, and the enzymatic activity from 2.5 to 5.0 U/g, while the pH did not show significant variations (Table 5).

Fig 8 shows the Pareto Diagram for droplet size (A), PDI (B), Zeta Potential (C) and enzyme activity (D). The particle size distribution analysis is an important parameter for evaluating the stability of a dispersed system, with the dynamic light scattering (DLS) technique being an accurate and suitable tool for investigating the internal properties of a microemulsified system [63]. As a result of the droplet size response (Fig 8A), the time and enzyme concentration variables did not significantly influence this parameter, only the interaction between these factors had a significant influence on the response. The statistical analysis for the PDI response (Fig 8B) showed that the enzyme concentration and time variables did not influence the responses.

Zeta Potential is a technique for determining the surface charge of particles in a colloidal solution with the aim of providing an estimate of the physicochemical stability of the analyzed system [68]. Therefore, high zeta potential values, whether positive or negative (more negative than -30 mV or more positive than +30 mV), reveal good physical-chemical stability due to the tendency for there to be repulsion between particles, avoiding aggregations from collisions between particles [64,65]. The results found regarding the zeta potential response are shown in the Pareto diagram (Fig 8C). It is possible to observe that the variables time (Linear term- L), enzyme concentration (Linear term) and the interaction between these presented statistical significance (p > 0.05). In analyzing the results, it is noted that the time and enzyme variables had a positive influence on the zeta

**Table 5. Central composite rotational design matrix (DCCR).**

| Assay | Lipase (%) | Agitation time (min.) | Droplet size (nm) | PDI | Zeta Potential (mV) | Enzyme Activity (U/g) |
|---|---|---|---|---|---|---|
| 1 | 10.00 (-1) | 5.0 (-1) | 1118.6 | 0,825 | -70.60 | 2.5 |
| 2 | 10.00 (-1) | 10.0 (+1) | 713.5 | 0,747 | -55.93 | 5.0 |
| 3 | 15.00 (+1) | 5.0 (-1) | 443.5 | 0,765 | -59.03 | 2.5 |
| 4 | 15.00 (+1) | 10.0 (+1) | 1181.3 | 0,900 | -58.66 | 5.0 |
| 5 | 8.96 (-1,41) | 7.5 (0) | 1274.3 | 0,893 | -65.50 | 5.0 |
| 6 | 16.03 (+1,41) | 7.5 (0) | 889.36 | 0,831 | -57.13 | 2.5 |
| 7 | 12.50 (0) | 4,0 (-1,41) | 1171 | 0,896 | -67.03 | 2.5 |
| 8 | 12.50 (0) | 11.0 (+1,41) | 809.7 | 0,704 | -59.73 | 5.0 |
| 9 (C) | 12.50 (0) | 7.5 (0) | 929.3 | 0,769 | -63.00 | 2.5 |
| 10(C) | 12.50 (0) | 7.5 (0) | 976 | 0,809 | -69.16 | 2.5 |

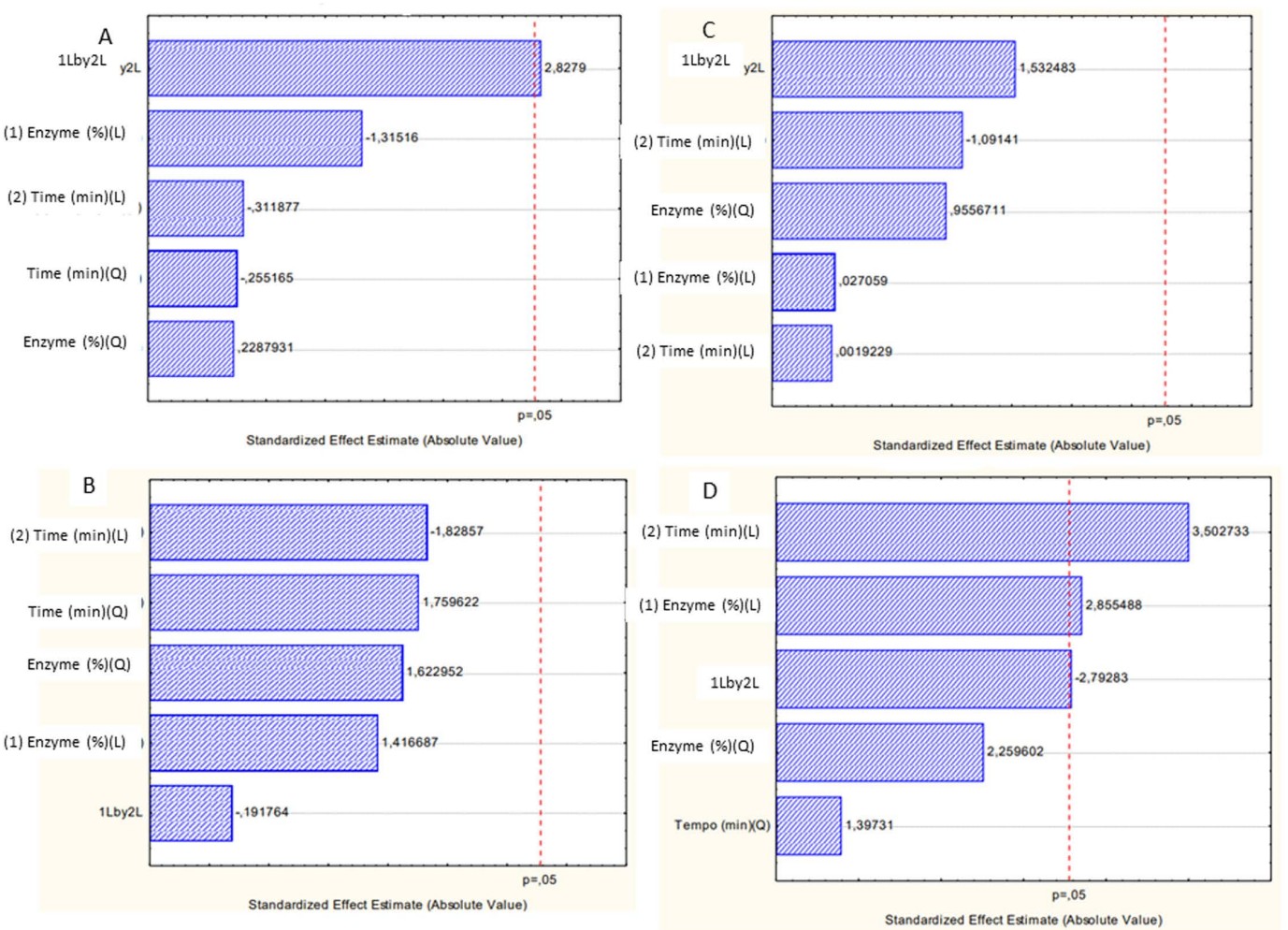

**Fig 8. Pareto diagram for response droplet size (A), PDI (B), pH (C) and Zeta Potential (D).**

potential response, which can be explained because this parameter can be affected by the intrinsic properties of the particles, such as size and concentration. Furthermore, an increase in temperature was observed during the preparation of cosmetic formulations when stirred for more than 5 minutes, which may have contributed to an increase in zeta potential, as suspension conditions such as pH, temperature and ionic strength can influence this result [66–68]. In a study conducted by Figueiredo and Campelo (2018) [68] in addressing the effect of process conditions on zeta potential values, an increase in the value of this parameter was also observed with longer homogenization time in preparing an emulsion containing a vegetable oil.

Only the stirring agitation time (linear term - L) showed a positive influence for the enzymatic activity response (Fig 8D). The influence of stirring time on enzymatic activity can be explained due to increased agitation increasing the homogeneity of the medium, causing the enzyme to be better dispersed throughout the formulation and thus providing an increase in enzymatic activity [69]. Other works also reference the positive influence of stirring: in analyzes carried out by Gawas, Khan and Rathod (2019) [70] using lipase to catalyze a synthesis reaction, it was observed that ultrasound stirring may have provided better solution mixing, which increased the interaction between the substrate and the enzyme.

Next, models were generated from the fitted regression coefficients obtained by the Statistica 7.0 software program which correlate the droplet size, zeta potential and enzymatic activity responses with the factors, as observed in the equations below:

$$Zeta\ Potential\ (mV) = -66.08 + 2.58\ x_1\ + 2.70\ x_1^2\ + 3.17\ x_2 + 1.67\ x_2^2 - 3.57\ x_1 x_2 \tag{4}$$

$$Enzyme\ Activity\ (U/g) = 2.50 - 0.44\ x_1\ + 0.62\ x_1^2\ + 1.07\ x_2 + 0.62\ x_2^2 \tag{5}$$

In which: $X_1$ is the lipase concentration (%) and $X_2$ is the stirring time (min.).

According to the analysis of variance (Table 6), the F values calculated for the Zeta Potential (Eq 4) and Enzyme Activity (Eq 5) models were greater than the respective F values tabulated in the 95% range reliable. Therefore, they can be considered statistically significant, as they presented coefficients of determination which suggest that the models are predictive.

According to the response surface graph in relation to the zeta potential response (Fig 9A), lower enzyme concentrations together with shorter agitation times cause an increase in the zeta potential (-70 mV). This is considered a satisfactory result, as this parameter is used to predict and control the formulation's stability, and the higher its value, the more likely the emulsion will be stable, because the charged particles repel each other and this force overcomes the natural tendency to aggregation [71,72].

The surface response graph regarding enzymatic activity (Fig 9B) shows that longer stirring times promote an increase in enzymatic activity. This condition was also addressed in a study evaluating different parameters for cellulase action on a polyester/cotton fabric, reporting that increased stirring increased enzymatic action [73,74]. At the end of the test, it was possible to observe that the cosmetic formulation presented a homogeneous appearance, as shown in Fig 10.

**Table 6. Analysis of variance for the models obtained.**

| Source of variation | Quadratic sum | Degrees of freedom | Quadratic mean | $F^*$ value | F tabulated (95%) |
|---|---|---|---|---|---|
| **Zeta potential (Eq 5)** | | | | | |
| Regression | 184.98 | 3 | 61.66 | 5.10 | $F_{3,6}$=4.76 |
| Residual | 72.47 | 6 | 12.08 | | |
| Total | 257.45 | 9 | | | |
| $R^2$=0,89349 | | | | | |
| **Enzyme activity (Eq 6)** | | | | | |
| Regression | 9.11 | 1 | 9.11 | 10.46 | $F_{1,8}$=5.32 |
| Residual | 6.96 | 8 | 0.87 | | |
| Total | 16.07 | 9 | | | |
| $R^2$=0,87796 | | | | | |

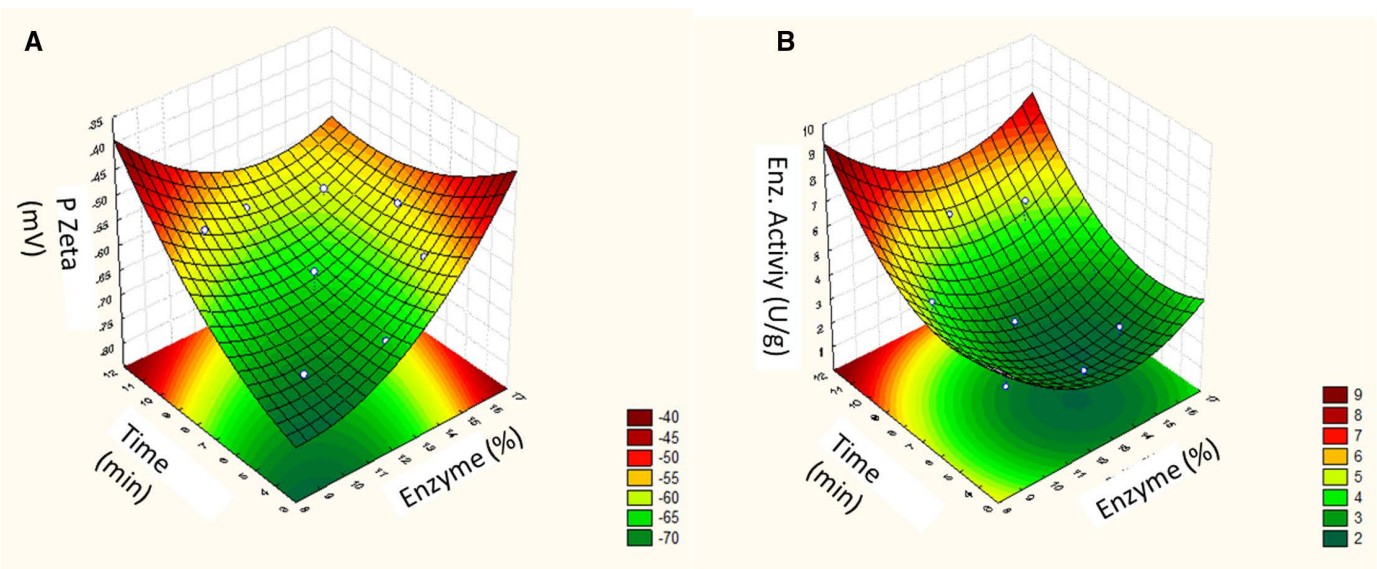

**Fig 9. Response surface for the dependent variable zeta potential (A) and enzymatic activity (B) as a function of time and enzyme concentration (%).**

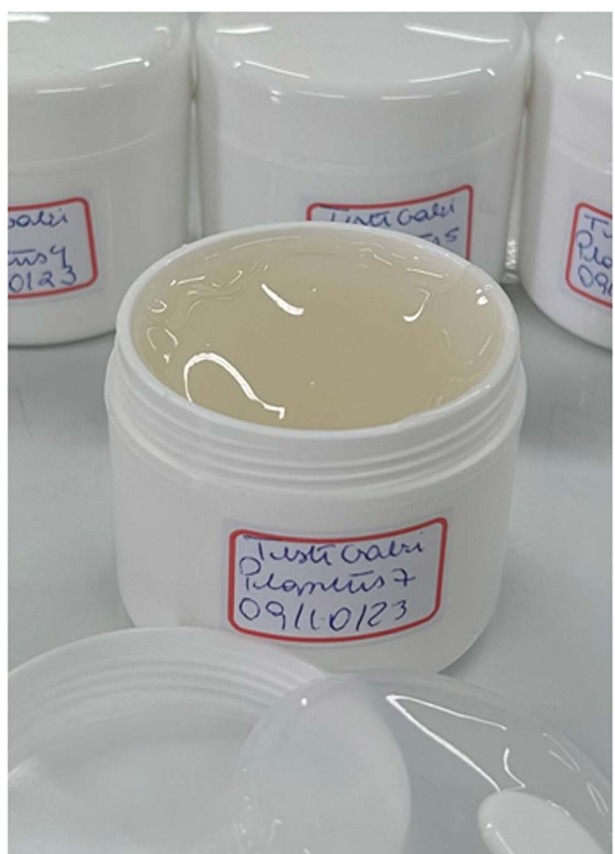

**Fig 10. Formulation containing lipase.**

## Conclusion

The lipase production from *Aspergillus terreus* showed satisfactory results of lipolytic activity in Solid State Fermentation (SSF) using wheat bran as substrate and olive oil as an inducing agent. The enzyme produced by *A. terreus* showed enzymatic activity values higher than those of the commercial lipase from *Rhizopus oryzae*. The enzymatic immobilization of the lipase from *A. terreus* showed a low yield, possibly due to a weak interaction between the enzyme and silica. The experimental design results showed that the variables agitation time and concentration of lipase from *A. terreus* influenced the zeta potential response. In contrast, only the agitation time variation was significant for enzymatic activity. The data found in the study show that the enzyme from *A. terreus* has potential for application in cosmetic formulations. This study is a pioneer in the use of a safe enzymatic crude extract in cosmetic formulations, which is necessary to guarantee the development of stable, safe and effective cosmetic products.

## Supporting information

**S1 Data. Dataset 1.**
(XLSX)

**S2 Data. Dataset 2.**
(XLSX)

**S3 Data. Dataset 3.**
(XLSX)

## Acknowledgments

The authors thank the LAMMEN – ECT/UFRN Laboratory for the XRD analyses.

## Author contributions

**Conceptualization:** Elissa Arantes Ostrosky.

**Formal analysis:** Gabriela Rocha Ramos.

**Investigation:** Gabriela Rocha Ramos.

**Methodology:** Gabriela Rocha Ramos, Patrícia Santos Lopes, Newton Andréo Filho, Lohanna Luciyanla Kakuda, Jéssica Nascimento da Silva Pinto, Neyna Santos Morais, Nathalia Saraiva Rios, Ana Paula Barreto Gomes, Francisco Humberto Xavier Júnior, Cristiane Fernandes de Assis.

**Project administration:** Cristiane Fernandes de Assis.

**Resources:** Gabriela Rocha Ramos, Francisco Canindé de Sousa Júnior.

**Software:** Emanuela de Lima Viana, Francisco Canindé de Sousa Júnior.

**Supervision:** Elissa Arantes Ostrosky, Cristiane Fernandes de Assis.

**Validation:** Francisco Canindé de Sousa Júnior.

**Writing – review & editing:** Cristiane Fernandes de Assis.

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
