## [Decision Letter · Decision Letter 0]

11 Sep 2024

PONE-D-24-30704Developing a cosmetic formulation containing lipase produced by the fungus Aspergillus terreusPLOS ONE

Dear Dr. de Assis,

Thank you for submitting your manuscript to PLOS ONE. After careful consideration, we feel that it has merit but does not fully meet PLOS ONE’s publication criteria as it currently stands. Therefore, we invite you to submit a revised version of the manuscript that addresses the points raised during the review process.

**ACADEMIC EDITOR: Please revise the work as per the comments specifically on methodology, introduction and conclusions sections.**

We look forward to receiving your revised manuscript.

Kind regards,

Amitava Mukherjee, ME, Ph.D.

Academic Editor

PLOS ONE

Journal requirements: 1. When submitting your revision, we need you to address these additional requirements.Please ensure that your manuscript meets PLOS ONE's style requirements, including those for file naming. The PLOS ONE style templates can be found at https://journals.plos.org/plosone/s/file?id=wjVg/PLOSOne_formatting_sample_main_body.pdf and https://journals.plos.org/plosone/s/file?id=ba62/PLOSOne_formatting_sample_title_authors_affiliations.pdf. 2. We note that the grant information you provided in the ‘Funding Information’ and ‘Financial Disclosure’ sections do not match. When you resubmit, please ensure that you provide the correct grant numbers for the awards you received for your study in the ‘Funding Information’ section.3. Thank you for stating the following financial disclosure:   [This work received financial support from the Coordenação de Aperfeiçoamento de Pessoal de Nível Superior, which granted the scholarship (001).].  Please state what role the funders took in the study.  If the funders had no role, please state: ""The funders had no role in study design, data collection and analysis, decision to publish, or preparation of the manuscript."" If this statement is not correct you must amend it as needed. Please include this amended Role of Funder statement in your cover letter; we will change the online submission form on your behalf. 4. We note that Figure 2 in your submission contain copyrighted images. All PLOS content is published under the Creative Commons Attribution License (CC BY 4.0), which means that the manuscript, images, and Supporting Information files will be freely available online, and any third party is permitted to access, download, copy, distribute, and use these materials in any way, even commercially, with proper attribution. For more information, see our copyright guidelines: http://journals.plos.org/plosone/s/licenses-and-copyright. We require you to either (1) present written permission from the copyright holder to publish these figures specifically under the CC BY 4.0 license, or (2) remove the figures from your submission: a. You may seek permission from the original copyright holder of Figure 2 to publish the content specifically under the CC BY 4.0 license.  We recommend that you contact the original copyright holder with the Content Permission Form (http://journals.plos.org/plosone/s/file?id=7c09/content-permission-form.pdf) and the following text:“I request permission for the open-access journal PLOS ONE to publish XXX under the Creative Commons Attribution License (CCAL) CC BY 4.0 (http://creativecommons.org/licenses/by/4.0/). Please be aware that this license allows unrestricted use and distribution, even commercially, by third parties. Please reply and provide explicit written permission to publish XXX under a CC BY license and complete the attached form.” Please upload the completed Content Permission Form or other proof of granted permissions as an ""Other"" file with your submission. In the figure caption of the copyrighted figure, please include the following text: “Reprinted from [ref] under a CC BY license, with permission from [name of publisher], original copyright [original copyright year].” b. If you are unable to obtain permission from the original copyright holder to publish these figures under the CC BY 4.0 license or if the copyright holder’s requirements are incompatible with the CC BY 4.0 license, please either i) remove the figure or ii) supply a replacement figure that complies with the CC BY 4.0 license. Please check copyright information on all replacement figures and update the figure caption with source information. If applicable, please specify in the figure caption text when a figure is similar but not identical to the original image and is therefore for illustrative purposes only. 5. Please include captions for your Supporting Information files at the end of your manuscript, and update any in-text citations to match accordingly. Please see our Supporting Information guidelines for more information: http://journals.plos.org/plosone/s/supporting-information. 

Reviewers' comments:

Reviewer's Responses to Questions

**Comments to the Author**

1. Is the manuscript technically sound, and do the data support the conclusions?

Reviewer #1: Partly

2. Has the statistical analysis been performed appropriately and rigorously? 

Reviewer #1: No

3. Have the authors made all data underlying the findings in their manuscript fully available?

Reviewer #1: No

4. Is the manuscript presented in an intelligible fashion and written in standard English?

Reviewer #1: No

5. Review Comments to the Author

Reviewer #1: The paper aimed to evaluate the potential of lipase from Aspergillus terreus (active ingredient) in cosmetic formulations. The enzymes were characterized using several instrumental techniques. Reviewing the paper I found several drawbacks and therefore I must say that the paper is not suitable for publication in Plos One.

Introduction section is poorly written and should be a little bit elaborative which will include scientific interests to the readers. In spite of sufficient data, instrumental techniques has not been well described/justified. Conclusion section is poorly written. Therefore I must reject it.

6. PLOS authors have the option to publish the peer review history of their article (what does this mean? ). If published, this will include your full peer review and any attached files.

**Do you want your identity to be public for this peer review?** For information about this choice, including consent withdrawal, please see our Privacy Policy .

Reviewer #1: No

---

## [Author Response · Author response to Decision Letter 1]

22 Sep 2024

RESPONSE TO REVIEWERS

1. When submitting your revision, we need you to address these additional requirements. Please ensure that your manuscript meets PLOS ONE's style requirements, including those for file naming. The PLOS ONE style templates can be found at https://journals.plos.org/plosone/s/file?id=wjVg/PLOSOne_formatting_sample_main_body.pdf and https://journals.plos.org/plosone/s/file?id=ba62/PLOSOne_formatting_sample_title_authors_affiliations.pdf.

We appreciate the comments. The suggestion was accepted

This study was partly financed by the Coordenação de Aperfeiçoamento de Pessoal de Nível Superior - Brasil (CAPES) - Finance Code 001.

3. Thank you for stating the following financial disclosure: This work received financial support from the Coordenação de Aperfeiçoamento de Pessoal de Nível Superior, which granted the scholarship (001).]. Please state what role the funders took in the study. If the funders had no role, please state: "The funders had no role in study design, data collection and analysis, decision to publish, or preparation of the manuscript. If this statement is not correct you must amend it as needed. Please include this amended Role of Funder statement in your cover letter; we will change the online submission form on your behalf.

4. We note that figure 2 in your submission contain copyrighted images. All PLOS content is published under the Creative Commons Attribution License (CC BY 4.0), which means that the manuscript, images, and Supporting Information files will be freely available online, and any third party is permitted to access, download, copy, distribute, and use these materials in any way, even commercially, with proper attribution. For more information, see our copyright guidelines: http://journals.plos.org/plosone/s/licenses-and-copyright. We require you to either (1) present written permission from the copyright holder to publish these figures specifically under the CC BY 4.0 license, or (2) remove the figures from your submission: a. You may seek permission from the original copyright holder of Figure 2 to publish the content specifically under the CC BY 4.0 license.

Figure 2 does not contain copyrighted images. Figure 2 contains Scanning Electron Microscopy images obtained by the Laboratory of Structural Characterization of Materials, Department of Materials Engineering – DEMat, Federal University of Rio Grande do Norte. The captions of Figure 2 have been removed not to generate doubts regarding copyright.

Reviewer #1:The paper aimed to evaluate the potential of lipase from Aspergillus terreus (active ingredient) in cosmetic formulations. The enzymes were characterized using several instrumental techniques. Reviewing the paper I found several drawbacks and therefore I must say that the paper is not suitable for publication in Plos One. Introduction section is poorly written and should be a little bit elaborative which will include scientific interests to the readers. In spite of sufficient data, instrumental techniques has not been well described/justified. Conclusion section is poorly written. Therefore I must reject it.

Thank you for considering our manuscript. We appreciate the comments and made careful reorganization of the content, considering all the points mentioned. We made the proper changes in the text highlighting in yellow

---

## [Decision Letter · Decision Letter 1]

16 Jan 2025

PONE-D-24-30704R1Developing a cosmetic formulation containing lipase produced by the fungus Aspergillus terreusPLOS ONE

Dear Dr. de Assis,

Thank you for submitting your manuscript to PLOS ONE. After careful consideration, we feel that it has merit but does not fully meet PLOS ONE’s publication criteria as it currently stands. Therefore, we invite you to submit a revised version of the manuscript that addresses the points raised during the review process.

We look forward to receiving your revised manuscript.

Kind regards,

Amitava Mukherjee, ME, Ph.D.

Academic Editor

PLOS ONE

Reviewers' comments:

Reviewer's Responses to Questions

**Comments to the Author**

1. If the authors have adequately addressed your comments raised in a previous round of review and you feel that this manuscript is now acceptable for publication, you may indicate that here to bypass the “Comments to the Author” section, enter your conflict of interest statement in the “Confidential to Editor” section, and submit your "Accept" recommendation.

Reviewer #2: All comments have been addressed

Reviewer #3: All comments have been addressed

2. Is the manuscript technically sound, and do the data support the conclusions?

Reviewer #2: Partly

Reviewer #3: Partly

3. Has the statistical analysis been performed appropriately and rigorously? 

Reviewer #2: Yes

Reviewer #3: No

4. Have the authors made all data underlying the findings in their manuscript fully available?

Reviewer #2: Yes

Reviewer #3: Yes

5. Is the manuscript presented in an intelligible fashion and written in standard English?

Reviewer #2: Yes

Reviewer #3: No

6. Review Comments to the Author

Reviewer #2: The article can be improved. I will leave some suggestions and some things that in my opinion need mandatory corrections.

1 -In Table 1, levels +1 and +1.41 have completely wrong values. Recalculate, as level +1 cannot be lower than level 0.

2- In line 357 is it really Figs?

3 - Likewise for line 369.

4 - Review the entire body of text. Why do authors abbreviate the word Figure?

5 - What is the purpose of XRD analysis? What information is taken from this analysis? Would there be a difference in the composition of the final product if the silica was crystalline?

5 - Improve the quality of Figures. In some cases, the values and/or words are not readable.

6 - The discussions after each Table are very superficial. The authors should discuss the results further, including justifying the differences obtained and why this occurred. Of course, it can be based on literature data.

7 - Is Table 6 really necessary? It would not be possible to just use the R2 Value in text form.

8 - Figure 10 is very bad. I suggest taking a photo of the cosmetic only, with a top view. In addition to what is written on 10/09/23.

9 - There is no bibliographic reference for the year 2024. However, I do not know if the study is up to date.

10 - From the year 2023, only 3 references. I suggest that the authors search the literature again to see if there really is nothing new, similar or the same to this study.

11- "This study is a pioneer in the use of a safe enzymatic crude extract in cosmetic formulations, which is necessary to guarantee the development of stable, safe and effective cosmetic products". Is it really? Was there a search for patents?

Reviewer #3: The article titled “Developing a cosmetic formulation containing lipase produced by the fungus Aspergillus terreus” investigates the development of a cosmetic formulation containing lipase produced by Aspergillus terreus.

Çalışma PLOSONE dergisine bilimsel olarak katkıda bulunabilir, ancak bazı sorunlar aşağıda listelenmiştir:

1. The introduction section in the manuscript is excessively long and the scientific novelty and originality addressed by the study are lacking in the explanation section.

2. Most importantly, why Aspergillus terreus was chosen and what advantages its lipase can offer in cosmetic formulations? It is not clear, it needs to be explained in detail.

3. The article lacks the rationale for choosing gel silica as a support material for immobilization. A comparison with alternative supports should be added.

4. Immobilization

5. The rationale for the design parameters for immobilization on the support material is unclear. The immobilization steps, such as the lipase enzyme concentrations and immobilization times, are not clear.

6. The statistical methods need to be clarified. It should be explained whether the assumptions for ANOVA (e.g. normality, homoscedasticity) were checked.

7. Enzyme immobilization results are poorly explained. Low yield (26.5%) should be discussed with more details on potential improvements in the method or support selection.

8. SEM and XRD analyses show that immobilization efficiency is low. It is clear that inadequate immobilization does not benefit the study. Immobilization parameters should be reviewed.

9. It is stated in the manuscript that immobilized lipase has a cytotoxic effect. These results contradict the aim of producing a safe product. It should be explained.

10. In addition, there is no fluency in the text and there are inconsistencies in the writing. The entire article should be corrected in this sense.

7. PLOS authors have the option to publish the peer review history of their article (what does this mean? ). If published, this will include your full peer review and any attached files.

**Do you want your identity to be public for this peer review?** For information about this choice, including consent withdrawal, please see our Privacy Policy .

Reviewer #2: **Yes: ** Marcelo Luis Mignoni

Reviewer #3: No

---

## [Author Response · Author response to Decision Letter 2]

25 Feb 2025

RESPONSE TO REVIEWERS

Reviewer #2: The article can be improved. I will leave some suggestions and some things that in my opinion need mandatory corrections.

1. In Table 1, levels +1 and +1.41 have completely wrong values. Recalculate, as level +1 cannot be lower than level 0.

Thank you for your careful review. We sincerely apologize for the error in Table 1, where the values for levels 0 and +1 were mistakenly swapped. We have corrected the table accordingly.

2. In line 357 is it really Figs?

We appreciate the comments and corrected the word figure.

3 - Likewise for line 369.

We appreciate the comments and corrected the word figure.

4 - Review the entire body of text. Why do authors abbreviate the word Figure?

We appreciate the comments and corrected the word figure.

5 - What is the purpose of XRD analysis? What information is taken from this analysis? Would there be a difference in the composition of the final product if the silica was crystalline?

The X-ray Diffraction (XRD) technique was used to evaluate the interactions between the enzyme and silica, specifically regarding the bonds formed. The results indicated that the crosslinking created an amorphous silica structure, meaning it preserved its initial form. Silica can exhibit polymorphism depending on the temperature and pressure applied in the system. In our study, since the temperature and pressure conditions were mild, the silica and silica maintained their original amorphous conformation as a support for lipase enzyme immobilization. The test was conducted to analyze the immobilized enzymes' physical properties and estimate their thermal stability since crystalline materials tend to be more thermally stable than amorphous materials (Ficanha et al., 2015).

5 - Improve the quality of Figures. In some cases, the values and/or words are not readable.

We appreciate the comments and adjusted all the figures to improve the resolution.

6 - The discussions after each Table are very superficial. The authors should discuss the results further, including justifying the differences obtained and why this occurred. Of course, it can be based on literature data.

We appreciate the comments and made the proper changes, highlighting them in yellow.

7 - Is Table 6 really necessary? It would not be possible to just use the R2 Value in text form.

The F test in Table 6 is indeed important for evaluating the overall significance of the model in the context of a Central Composite Rotational Design (CCRD). While the R² value provides an indication of how well the model fits the data, it does not tell us whether the relationship between the predictors and the response variable is statistically significant.

The F test provides a more rigorous validation of the model, ensuring that any conclusions drawn from the regression analysis are not based on random chance or noise in the data. In experimental designs like CCRD, this validation is crucial, especially when optimizing processes or making predictions. Thus, while R² offers insight into model fit, the F test is essential for confirming the model's significance, which justifies its inclusion in Table 6. Without the F test, it would be difficult to assess whether the observed relationships between factors and responses are statistically reliable.

8 - Figure 10 is very bad. I suggest taking a photo of the cosmetic only, with a top view. In addition to what is written on 10/09/23.

We appreciate the comments and adjusted all the figures to change the figure.

9 - There is no bibliographic reference for the year 2024. However, I do not know if the study is up to date.

Os artigos referente a lipase em cosméticos são bem escassos e não recentes. Não encontramos nas bases de dados artigos de 2024 referentes a lipase e cosméticos.

10 - From the year 2023, only 3 references. I suggest that the authors search the literature again to see if there really is nothing new, similar or the same to this study.

Realmente não existem trabalhos recentes referente a esse tema sendo possível conseguir apenas alguns números de patentes que trazem poucas informações sobre o tema.

11- "This study is a pioneer in the use of a safe enzymatic crude extract in cosmetic formulations, which is necessary to guarantee the development of stable, safe and effective cosmetic products". Is it really? Was there a search for patents?

We conducted a patent search and found some that were related exclusively to the enzyme lipase and others where lipase is combined with other enzymes in the product. However, the information found is quite general and does not provide enough detail to support a thorough discussion of the results. Patent EP 0097810 A1 describes a shampoo with stable lipolytic enzymatic activity over time, which contains as essential components a non-ionic surfactant (chosen from fatty acid esters with polyalcohols, ethoxylated or not, fatty acids, and ethoxylated alkylphenols) and a pancreatic, plant, or fungal type lipase. Patent EP 0713696 A1 describes a topical product capable of releasing hydroxy acids onto the skin, using lipase as the enzyme. Patent U.S 6153205 A presents a topical product for the skin containing lipase, a vitamin precursor used in cosmetics and/or dermatology, and alcohol. Additionally, patent WO 0076458 A3 discusses the application of lipase, protease, and peroxidase in cosmetic products with regenerative action on the skin. Patent JP2004244355-A describes a cosmetic formulation for preventing and treating localized fat, combining caffeine and lipase. The use of enzymes in cosmetic products with actions beyond lipolysis is also mentioned in other patents. Patent U.S 6514506 B1 describes a whitening composition with an enzymatic extract of melanase for removing skin stains. Patent WO 2006/018048 A1 addresses the stability of aqueous (cyclodextrins) and non-aqueous enzymatic release systems in cosmetic products containing lipases, proteases, and oxidases. Finally, patent U.S. 20030026794 A1 describes a method for skin treatment with selective enzymes (proteases) that remove specific layers of the skin.

Reviewer #3: The article titled “Developing a cosmetic formulation containing lipase produced by the fungus Aspergillus terreus” investigates the development of a cosmetic formulation containing lipase produced by Aspergillus terreus.

1. The introduction section in the manuscript is excessively long and the scientific novelty and originality addressed by the study are lacking in the explanation section.

We appreciate the comments and made the proper changes, highlighting them in yellow.

2. Most importantly, why Aspergillus terreus was chosen and what advantages its lipase can offer in cosmetic formulations? It is not clear, it needs to be explained in detail.

We appreciate the comments and added information on pages 1 and lines 96-98 regarding the choice of the fungus Aspergillus terreus. Our group has been dedicated to lipase production, as demonstrated in the works by (de Azevedo et al., 2020)and (Barros et al., 2023), where the enzyme showed high activity. In conjunction with biotechnology, the cosmetic industry has been using enzymes for various topical applications, such as antioxidants, proteases, DNA repair enzymes, and specifically lipases, which are used in shampoos, skin cleansing, and even in treating cellulite. With the experience gained in lipase production, we are now beginning a study on cosmetic formulations to identify the best conditions to maintain the enzyme's stability and activity, aiming for its application in future formulations.

3. The article lacks the rationale for choosing gel silica as a support material for immobilization. A comparison with alternative supports should be added.

We appreciate the comments and added alternative supports to the discussion. We made the proper change in the text, highlighting it in yellow.

4. Imobilization

Silica immobilization has been extensively studied and has shown promising results for lipase immobilization, depending on the application for which the enzyme will be used. However, some reports show that these results have been minimal because they do not show the necessary mechanical stability.

5. The rationale for the design parameters for immobilization on the support material is unclear. The immobilization steps, such as the lipase enzyme concentrations and immobilization times, are not clear.

We appreciate the comments and added detailed information about immobilization.

6. The statistical methods need to be clarified. It should be explained whether the assumptions for ANOVA (e.g. normality, homoscedasticity) were checked.

We appreciate the comments and detail the methodology used in statistics. We made the proper change in the text, highlighting it in yellow

7. Enzyme immobilization results are poorly explained. Low yield (26.5%) should be discussed with more details on potential improvements in the method or support selection.

We appreciate the comments and improved discussions on low immobilization efficiency. We made the proper change in the text, highlighting it in yellow

8. SEM and XRD analyses show that immobilization efficiency is low. It is clear that inadequate immobilization does not benefit the study. Immobilization parameters should be reviewed.

We appreciate the comments. This was the first support used for enzyme immobilization, and we observed that the result was unsatisfactory. However, we found it relevant to present these results since silica is a support used for lipase immobilization (Pedro et al., 2025) for other purposes. Still, this support was not suitable for use in cosmetics. We are developing other works using natural supports that can thus be used in cosmetics.

9. It is stated in the manuscript that immobilized lipase has a cytotoxic effect. These results contradict the aim of producing a safe product. It should be explained.

The work presents the results of the immobilized enzyme (with low immobilization efficiency) and the free enzyme applied to cosmetics. Although the immobilization efficiency was unsatisfactory, we decided, before testing new immobilization conditions, to evaluate the safety of the immobilized enzyme. During this evaluation, we observed that the silica used as a support was cytotoxic, and the enzyme immobilized on it. These results indicate that, besides the low immobilization efficiency, the process is not safe from a cellular perspective. From then on, we used the free enzyme (without immobilization) in the cosmetics, as this enzyme did not show cytotoxicity in the tested cells. The free enzyme was then added to the cosmetic, and an experimental plan was developed to evaluate the best application conditions for this enzyme and to verify whether the enzymatic activity was maintained.

10. In addition, there is no fluency in the text and there are inconsistencies in the writing. The entire article should be corrected in this sense.

Thank you for considering our manuscript. We made the proper change in the text, highlighting it in yellow. We have carefully addressed the manuscript's grammar, usage, and overall readability. A professional in English has revised the manuscript, and we also used the software Grammarly (full version) to check for gross mistakes further. Hopefully, this is a better version to revise.

References:

Barros, K. dos S., Assis, C. F. de, Jácome, M. C. de M. B., Azevedo, W. M. de, Ramalho, A. M. Z., Santos, E. S. dos, Passos, T. S., Sousa Junior, F. C. de, & Damasceno, K. S. F. da S. C. (2023). Bati Butter as a Potential Substrate for Lipase Production by Aspergillus terreus NRRL-255. Foods, 12(3), 564. https://doi.org/10.3390/foods12030564

de Azevedo, W. M., de Oliveira, L. F. R., Alcântara, M. A., Cordeiro, A. M. T. de M., Damasceno, K. S. F. da S. C., Assis, C. F. de, & Sousa Junior, F. C. de. (2020). Turning cacay butter and wheat bran into substrate for lipase production by Aspergillus terreus NRRL-255. Preparative Biochemistry and Biotechnology, 50(7), 689–696. https://doi.org/10.1080/10826068.2020.1728698

Ficanha, A. M. M., Nyari, N. L. D., Levandoski, K., Mignoni, M. L., & Dallago, R. M. (2015). Study of immobilization of lipase in silica by the sol-gel technique. Quimica Nova, 38(3), 364–369. https://doi.org/10.5935/0100-4042.20150027

Pedro, K. C. N. R., da Silva, G. A. R., da Silva, M. A. P., Henriques, C. A., & Langone, M. A. P. (2025). Immobilization of lipase on zeolite, silica, and silica-aluminas and its use in hydrolysis, esterification, and transesterification reactions. Catalysis Today, 447(October 2024). https://doi.org/10.1016/j.cattod.2024.115141

---

## [Decision Letter · Decision Letter 2]

18 Mar 2025

Developing a cosmetic formulation containing lipase produced by the fungus Aspergillus terreus

PONE-D-24-30704R2

Dear Dr. de Assis,

We’re pleased to inform you that your manuscript has been judged scientifically suitable for publication and will be formally accepted for publication once it meets all outstanding technical requirements.

Kind regards,

Amitava Mukherjee, ME, Ph.D.

Academic Editor

PLOS ONE

Additional Editor Comments (optional):

Reviewers' comments:

Reviewer's Responses to Questions

**Comments to the Author**

1. If the authors have adequately addressed your comments raised in a previous round of review and you feel that this manuscript is now acceptable for publication, you may indicate that here to bypass the “Comments to the Author” section, enter your conflict of interest statement in the “Confidential to Editor” section, and submit your "Accept" recommendation.

Reviewer #3: All comments have been addressed

2. Is the manuscript technically sound, and do the data support the conclusions?

Reviewer #3: Yes

3. Has the statistical analysis been performed appropriately and rigorously? 

Reviewer #3: Yes

4. Have the authors made all data underlying the findings in their manuscript fully available?

Reviewer #3: Yes

5. Is the manuscript presented in an intelligible fashion and written in standard English?

Reviewer #3: Yes

6. Review Comments to the Author

Reviewer #3: Dear Editor,

Major corrections noted in the article have been addressed. The article will contribute to Plos One. It is acceptable for publication.

Best regards.

7. PLOS authors have the option to publish the peer review history of their article (what does this mean? ). If published, this will include your full peer review and any attached files.

**Do you want your identity to be public for this peer review?** For information about this choice, including consent withdrawal, please see our Privacy Policy .

Reviewer #3: No

---

## [Editor Report · Acceptance letter]

PONE-D-24-30704R2

PLOS ONE

Dear Dr. de Assis,

I'm pleased to inform you that your manuscript has been deemed suitable for publication in PLOS ONE. Congratulations! Your manuscript is now being handed over to our production team.

Kind regards,

on behalf of

Professor Dr. Amitava Mukherjee

Academic Editor

PLOS ONE